# Diverse protein assembly driven by metal and chelating amino acids with selectivity and tunability

Minwoo Yang [1] & Woon Ju Song [1]*

Proteins are versatile natural building blocks with highly complex and multifunctional architectures, and self-assembled protein structures have been created by the introduction of covalent, noncovalent, or metal-coordination bonding. Here, we report the robust, selective, and reversible metal coordination properties of unnatural chelating amino acids as the sufficient and dominant driving force for diverse protein self-assembly. Bipyridine-alanine is genetically incorporated into a $D_3$ homohexamer. Depending on the position of the unnatural amino acid, 1-directional, crystalline and noncrystalline 2-directional, combinatory, and hierarchical architectures are effectively created upon the addition of metal ions. The length and shape of the structures is tunable by altering conditions related to thermodynamics and kinetics of metal-coordination and subsequent reactions. The crystalline 1-directional and 2-directional biomaterials retain their native enzymatic activities with increased thermal stability, suggesting that introducing chelating ligands provides a specific chemical basis to synthesize diverse protein-based functional materials while retaining their native structures and functions.

---

[1] Department of Chemistry, College of Natural Sciences, Seoul National University, Seoul 08826, Republic of Korea. *email: woonjusong@snu.ac.kr

The design and synthesis of supramolecular structures has been one of the primary interests at all scales[1–3]. Nature utilizes diverse biological molecules, such as DNA, peptides, and proteins as building blocks to create various architectures with high structural and functional complexity. An increasing number of artificial protein-assemblies has been also been reported[2,4–8]. Extensive sequence optimization creates electrostatic and hydrophobic interactions with size and shape complementarity, yielding novel protein–protein interfaces[9–13]. Covalent disulfide bonds[14,15] and native non-covalent interactions with DNA[16,17], peptides[18], proteins[19,20], or small molecules, such as heme, biotin, and sugar molecules[21–29] can also induce protein oligomerization, yielding self-assembled protein structures. Metal-coordination either to surface-exposed natural amino acids such as histidine and glutamate or to chemically modified amino acids[30–35] also endows strong driving force for self-assembly, while often requiring sequence optimization to create selective metal-binding sites[30,32]. While these methods have enabled the formation of numerous protein-assemblies, alternative and orthogonal approaches that are less dependent on the identity of the building blocks may expand the scope and diversity of the protein architectures.

Because multidentate chelating ligands exhibit considerably higher binding affinities for metal ions than several monodentate ligands combined (termed the chelation effect)[36], chelating amino acids may constitute more potent and selective protein–protein interactions than monodentate natural amino acids. Chelating amino acids have been applied to protein self-assembly[27,37,38], but only to the formation of discrete sized oligomers or to those aided by synthetic modifications. Herein, we genetically incorporate bidentate bipyridyl-alanine (bpy-Ala or bpy) into a protein for the formation of [M(bpy)2], where each bipyridine originates from a discrete protein (Fig. 1a). The polymerization driven by metal coordination to chelating ligands is explored with protein building blocks of which the sequence is neither optimized nor templated for self-assembly. The directionality of protein polymerization is also modulated to demonstrate that diverse protein architectures, one-directional (1D), crystalline and noncrystalline two-directional (2D), combinatory, and hierarchical structures, can be obtained. Thermodynamic and kinetic controls associated with inorganic reactivity are also applied to investigate whether the shape and length of the protein-assembled architectures can be tunable.

## Results

**Experimental design.** We propose that the selective formation of [M(bpy)2] can be a primary driving force for protein assembly, and therefore, generation of alternative species such as [M(bpy)3] and [M(bpy)L] should be repressed, where L represents a buffer-derived species, such as $OH^-$, $Cl^-$, $H_2O$, and surface-exposed metal-ligating residues. The formation of [M(bpy)3] can be suppressed by the steric hindrance of proteins, whereas the formation of ternary complexes such as [M(bpy)L] might prevail with high concentrations of ligands L and steric hindrance between proteins[39]. To resolve these issues, the thermodynamic parameters of the first-row transition metal ions such as $\log K_{OH^-}$, $\log K_1$, and $\log K_2$ for bpy ligands have been obtained from the literature[40,41] (Supplementary Table 1 and Supplementary Note 1). Metal ions with low $\log K_{OH^-}$ values are desirable because they represent the degrees of potential side-reactions between metal ions and $OH^-$ or acidic residues. In addition, the $\log K_1$ and $\log K_2$ values should be high and comparable to each other for the formation of [M(bpy)2]. In this regard, $Ni^{2+}$ satisfies the conditions. Notably, even though $Cu^{2+}$ exhibits a high binding affinity for bpy, as illustrated in the Irving–William series, we surmised that the relatively low $\log K_2$ relative to $\log K_1$ inhibits the formation of $[Cu(bpy)_2]^{2+}$-like species. In addition, for more selective $Ni^{2+}$ to bpy binding, we added an excess of salts, such as NaCl and LiCl, to suppress undesirable reactions between $Ni^{2+}$ and acidic residues by the formation of electrostatic interactions between the adventitious ligands and hard acidic cations, $Na^+$ and $Li^+$, instead[42].

Because our strategy requires neither sequence optimization nor noncovalent interactions with the native substrates or ligands, we surmise that there is little restriction in the choice of building blocks. Therefore, we selected a dihedral $D_3$ homohexamer, whose symmetry is less feasible for interfacial redesign but suitable for multidirectional crystalline protein-assembly. Notably, although a $D_3$ protein was recently applied for noncrystalline protein architectures, to the best of our knowledge, the protein symmetry has not been applied to the formation of crystalline-materials, expanding the diversity of protein building blocks. Among $D_3$ proteins deposited in RCSB, we excluded native metalloproteins to avoid the complexity associated with preexisting metal-binding sites and searched for proteins featuring surface-exposed, flexible domains to anchor the bpy ligands.

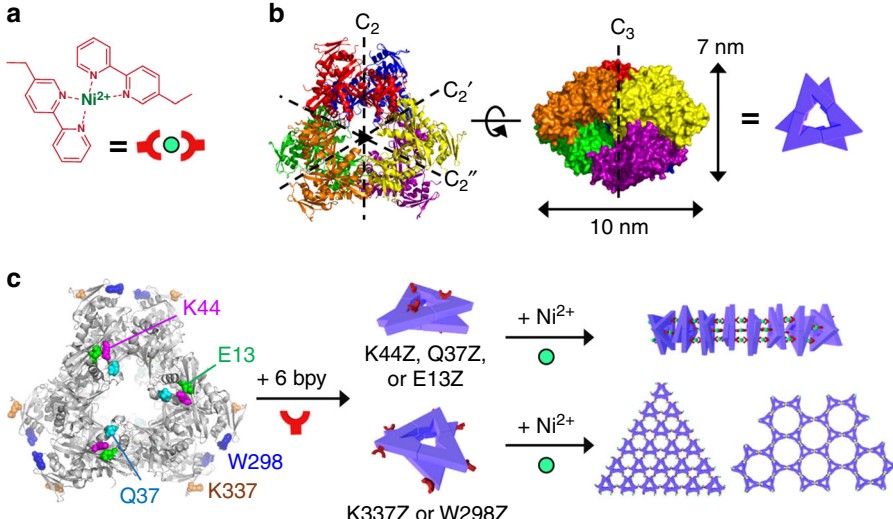

**Fig. 1** Symmetry-adapted protein assembly with metal-chelating amino acids. **a** $[Ni(bpy)_2]^{2+}$ formation as the primary driving force. **b** X-ray crystal structure of an acetyltransferase (PDB code 3N7Z). **c** A scheme for 1D assembly with K44Z, Q37Z, or E13Z (top) and 2D assembly using K337Z or W298Z (bottom) in the p312 (left) or p622 (right) layer symmetries.

Finally, acetyltransferase from *Bacillus anthracis*[43] was selected as a building block (Fig. 1b).

For 1D assembly, repeated linkages between the head and/or tail domains of the building blocks are required. Therefore, bipyridyl groups were introduced to K44, Q37, or E13 located in the top/bottom planes. 2D materials were designed by incorporating bipyridyl groups into K337 or W298 on the lateral faces (Fig. 1c). As possible 2D-layer symmetries, p312 and p622 spatial arrangements are supposedly possible when all bpy groups in the $D_3$ molecules participate in the formation of [Ni(bpy)$_2$] species by $C_2$ operation[44,45].

Using amber codon and orthogonal aminoacyl-tRNA synthases/tRNA pairs[46,47], bpy-Ala (Z) was incorporated into the selected sites by site-directed mutagenesis (Supplementary Table 2). We initially designed more variants than listed above (Supplementary Table 3), but not all were expressed on a large scale. All isolated proteins were treated with excess amounts of EDTA and prepared as a hexamer, retaining their native oligomeric state.

## 1D linear assemblies with K44Z

For the 1D linear assemblies, [Ni(bpy)$_2$] formation requires the addition of 3 equivalents of Ni$^{2+}$ to 1 equivalent of hexameric protein harboring 6 bpy ligands. The stoichiometric amounts of Ni$^{2+}$ were added to the K44Z protein, resulting in neither a vivid change in transparency nor an aggregates-like species. Negatively stained transmission electron microscope (TEM) images of the solution demonstrated that linear rod-shaped materials are formed with the repeating units, whose diameter and height correspond to the width and height of an individual protein, 10 and 7 nm, respectively (Fig. 2a). AFM images also revealed that the height of the 1D-assembled materials is consistent with the size of a hexamer (Fig. 2b). We manually measured and counted approximately 300 protein-rods, and the average ($L_n$) and maximum lengths were ~0.28 and 1.6 μm, respectively with a high polydispersity index (PDI) to be 2.08 (Fig. 2c, Supplementary Figs. 13 and 14).

To demonstrate that the linear protein-assembly is created under thermodynamic control such as the binding affinity of metal ions to bpy ligands, we replaced Ni$^{2+}$ with Co$^{2+}$, Cu$^{2+}$, and Zn$^{2+}$, which exhibit log $K_1$ or log ($K_2/K_1$) values that are not satisfactory for the formation of [M(bpy)$_2$] species. We also lowered the NaCl or LiCl concentrations for the reaction with Ni$^{2+}$ to vary the degree of masking electrostatic interactions (Supplementary Figs. 9 and 10). In all cases, no or considerably shorter rod-shaped materials were produced, indicating that the appropriate metal ions and salts are essential for the selective formation of [Ni(bpy)$_2$] and the consequent protein assembly. Alternatively, when we added one of the other metal ions listed above together or after Ni$^{2+}$, substantially shorter rods were formed compared with those formed with Ni$^{2+}$ alone, suggesting that the reactions are reversible and thermodynamically determined. When the strong metal chelator EDTA was added to the preformed rods, the length of the linear assembly shortened, indicating that [Ni(bpy)$_2$] functions as an effective and reversible connecting node for self-assembly. It is noteworthy that the connector, [Ni(bpy)$_2$], may possess additional ligands, yielding a [Ni(bpy)$_2$L]-like species. Regardless, the data described above definitively indicate that the reaction between Ni$^{2+}$ and two bpy molecules from discrete macromolecules enables the 1D self-assembly.

## Optimization of the linear assemblies

To tune and explore metal-dependent 1D-polymerization by altering the thermodynamic and/or kinetic factors operative in the reactions, various metal to protein stoichiometries, temperatures, and protein concentrations were applied. At 4 °C, the length of the linear assemblies was maximized when the amount of Ni$^{2+}$ reached the stoichiometric ratio (Fig. 3a and Supplementary Fig. 12). The use of greater than 3 equiv. of Ni$^{2+}$ to 1 equiv. of hexamer yields substantially shorter linear assemblies presumably because the excess Ni$^{2+}$ shifts the equilibrium from [Ni(bpy)$_2$] to [Ni(bpy)]-like species. The analogous metal-dependent length control also occurred at 22 and 37 °C, although the optimal concentrations of Ni$^{2+}$ gradually shifted from stoichiometric amounts to three to six and eightfold, respectively (Supplementary Figs. 13–17). These results suggest that the elevated temperature shifts the thermodynamic equilibrium or alters the rates of chemical reactions related to the formation of [Ni(bpy)$_2$] and concurrent polymerization.

Regardless of the origin of the temperature-dependent length control, the protein rods were shortened again when a concentration greater than the optimal of Ni$^{2+}$ was added at the subjected conditions, suggesting that thermodynamic control is still operative as described above. Notably, no aggregate was observed even with up to 30 equiv. of Ni$^{2+}$ to 1 equiv. of hexamer (Supplementary Figs. 13–16), possibly because Ni$^{2+}$ exhibits a relatively low $K_{OH}-$ value, and is thus less likely to react with surface-exposed acidic residues in a nonselective manner.

Because the pH value of Tris buffer is known to vary according to the applied temperature[48], we also examined whether the shift of the optimal Ni$^{2+}$ concentration is related to the altered pH rather than the temperature. The linear assemblies were generated

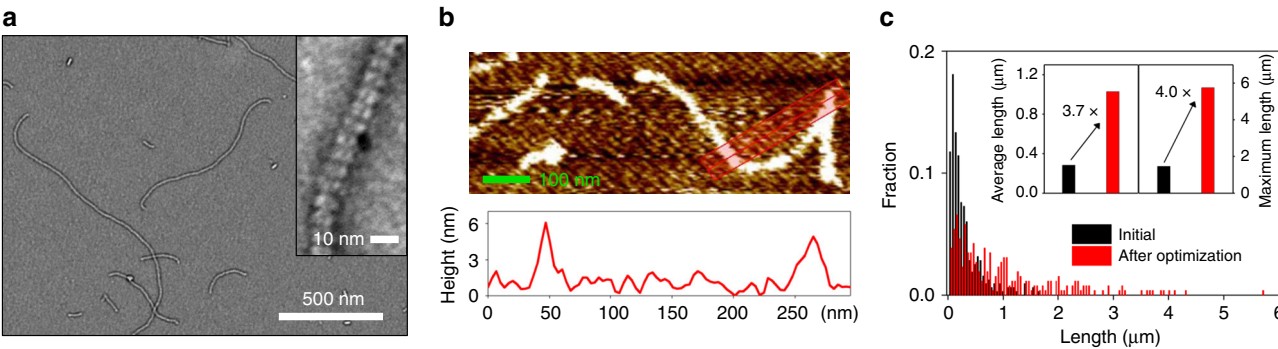

**Fig. 2** Formation of K44Z-rods. **a** TEM and **b** AFM images of K44Z (10 μM) incubated with Ni$^{2+}$ (3 equiv.) for 24 h at 22 °C. **c** The length in Fig. 2a (black) and after optimization (red); K44Z (50 μM) and 8 equiv. Ni$^{2+}$ (8 equiv.) at 37 °C. The average lengths ($L_n$) were determined by measuring 315 (PDI = 2.08) and 258 (PDI = 1.89) rods formed at the initial condition and after optimization, respectively. The raw data in Fig. 2c are provided as a Source Data file.

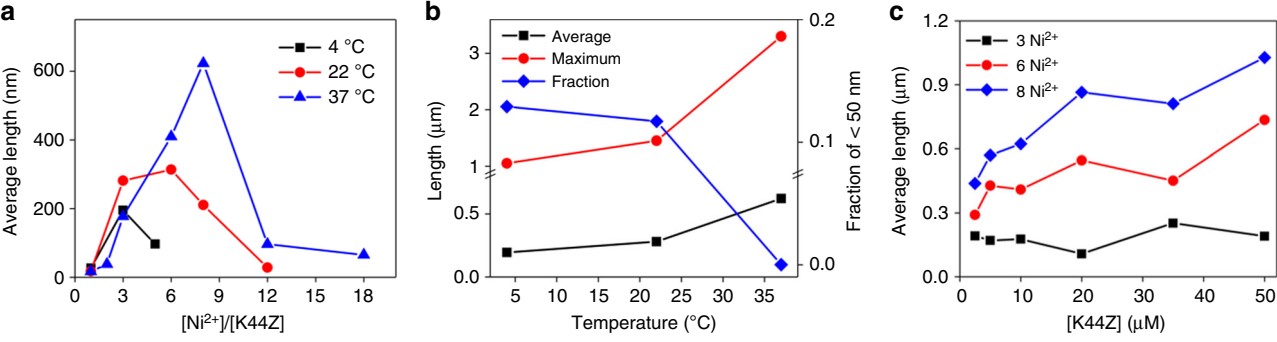

**Fig. 3** Optimization of K44Z-rod formation. **a** Various ratios of $Ni^{2+}$ to K44Z (10 μM) at 4–37 °C. **b** Reaction of K44Z (10 μM) and the optimal $Ni^{2+}$ at a given temperature. **c** Variations in the protein concentrations at 37 °C. The raw data in Fig. 3 are provided as a Source Data file.

even in various pH 6–9 buffers at 37 °C (Supplementary Fig. 27), but their lengths were considerably shorter than at pH 7. The results in pH 6 buffer are likely to be ascribed to the intrinsic protein instability, and the formation of protein aggregates even in the absence of metal ions, whereas the assemblies in pH 8–9 buffers are attributed to the higher concentrations of inhibitory ligands such as $OH^-$. Notably, the optimal $Ni^{2+}$ to protein ratio remains unaltered at all applied pH ranges, implicating that the optimal metal concentration is dependent on the temperatures rather than the pH value.

With the optimized concentrations of $Ni^{2+}$, the consumption of shorter than 50 nm long precursor-like species and the subsequent elongation of the assembled rods were enhanced at higher temperatures (Fig. 3b). These results suggest that although the initial binding of the metal to bpy is exothermic, the subsequent bimolecular reaction between two [Ni(bpy)] species is likely to be thermodynamically favorable at higher temperatures, possibly due to the elevated mobility of the macromolecules. The release of $Ni^{2+}$ might also become critical for the formation of $[Ni(bpy)_2]$, and this is likely to be endothermic. In addition, entropic effects might influence the self-assembly of macromolecules, often releasing hydrated water molecules by creating protein–protein interfaces[49]. We further elevated the temperatures to 43 and 49 °C, and linear assemblies were formed with no hint of aggregation or fibrillation (Supplementary Fig. 26). Their lengths, however, were substantially shorter than expected even with the optimal $Ni^{2+}$ concentration, which is presumably related to the intrinsic instability of the protein above 37 °C (Supplementary Fig. 25).

The length of the rods was also dependent on the protein concentration (Fig. 3c and Supplementary Figs. 18–24). With close to the optimal ratio (6–8 equiv.) of $Ni^{2+}$ to hexameric protein at 37 °C, the linear-assembled products extended up to ~2-fold upon variations of the protein concentration, indicating that the self-assembly is dependent on bimolecular reactions between the macromolecules. However, this dependence was not observed with the suboptimal concentrations of $Ni^{2+}$ (3 equiv.), implying that at a given condition, the protein concentration is not a limiting factor. Finally, when all conditions listed above were optimized, the K44Z protein assembly was detected to be 1.0 μm on average and up to ~5.8 μm as the maximum length, while the diameter of the 1D-materials corresponded to the size of the protein (Fig. 2c). These data demonstrate that chelating ligands can drive the formation of the 1D protein-assembly in predictable and tunable manners.

**Kinetics of the 1D rod formation**. To further explore whether 1D protein-assembly is under kinetic control, the $Ni^{2+}$ coordination to the bipyridyl ligand at the K44Z position was monitored

by time-dependent ultraviolet–visible (UV–vis) spectroscopy, size exclusion chromatography, and TEM. First, the absorption changes at 280 and 312 nm were monitored by the addition of $Ni^{2+}$ to K44Z with an isosbestic point at 295 nm (Fig. 4a), indicative of metal coordination to the bpy ligand[50]. The spectral changes were completed in ~20 s. Then, the concurrent changes in the molecular weight of the assembled species were monitored by size exclusion chromatography (Fig. 4b). Incubation of K44Z with $Ni^{2+}$ for 6 h converted all proteins into higher molecular weight species, which eluted at 40 mL. The polymerization process proceeded for 24 h, as illustrated by continuing changes in the elution profile. Time-dependent TEM images also indicated that the reaction continued for 24 h (Fig. 4c and Supplementary Figs. 28–31). Therefore, these kinetic data indicate that a slow elongation step proceeds after the rapid reactions between $Ni^{2+}$ and bpy. It is noteworthy that the rods grew substantially slowly upon the addition of 8 equiv. of $Ni^{2+}$ compared with 6 equiv. of $Ni^{2+}$ to 1 equiv. of hexamer, whereas the former condition resulted in longer 1D-materials. Therefore, these data imply that $Ni^{2+}$ release from the bimolecular reaction of [Ni(bpy)] species is involved as a rate-determining step, and slow conversion of [Ni(bpy)] to $[Ni(bpy)_2]$ is likely to be beneficial to synthesize the thermodynamically driven products. Then, the temperature-dependent optimal ratios of $Ni^{2+}$ to protein described above may originate from the orchestration of both thermodynamic and kinetic controls. With these observations, 1D assembly is likely to operate by isodesmic polymerization when all bipyridyl groups react with $Ni^{2+}$ simultaneously, and it is analogous to the formation of linear supramolecular polymers[51,52], where both thermodynamic and kinetic controls are operatively derived from the inorganic reactivity (Fig. 4d).

**Protein 1D-assembly with other variants**. To demonstrate that our strategy of anchoring a bpy ligand is not limited to a specific residue while metal coordination to the bpy ligand is dependent on the local environment, other than K44Z variants were designed for 1D-assembly. The reactions of Q37Z and E13Z with $Ni^{2+}$ were monitored by TEM (Fig. 1c). Q37Z generated aggregates, whereas E13Z formed linearly assembled products similar to, but substantially shorter than, those formed with K44Z (Supplementary Fig. 35). The discrete properties of the products are presumably related to the location of the residues. Q37Z is located in a slightly concave region, potentially creating substantial steric hindrance for protein–protein interactions. E13Z is located near the positively charged residue K44, which may perturb the formation of $[Ni(bpy)_2]$ (Fig. 5a and Supplementary Fig. 32).

To explore the coordination environment of E13Z, the E13Z/ K44E mutant was prepared by site-directed mutagenesis. The

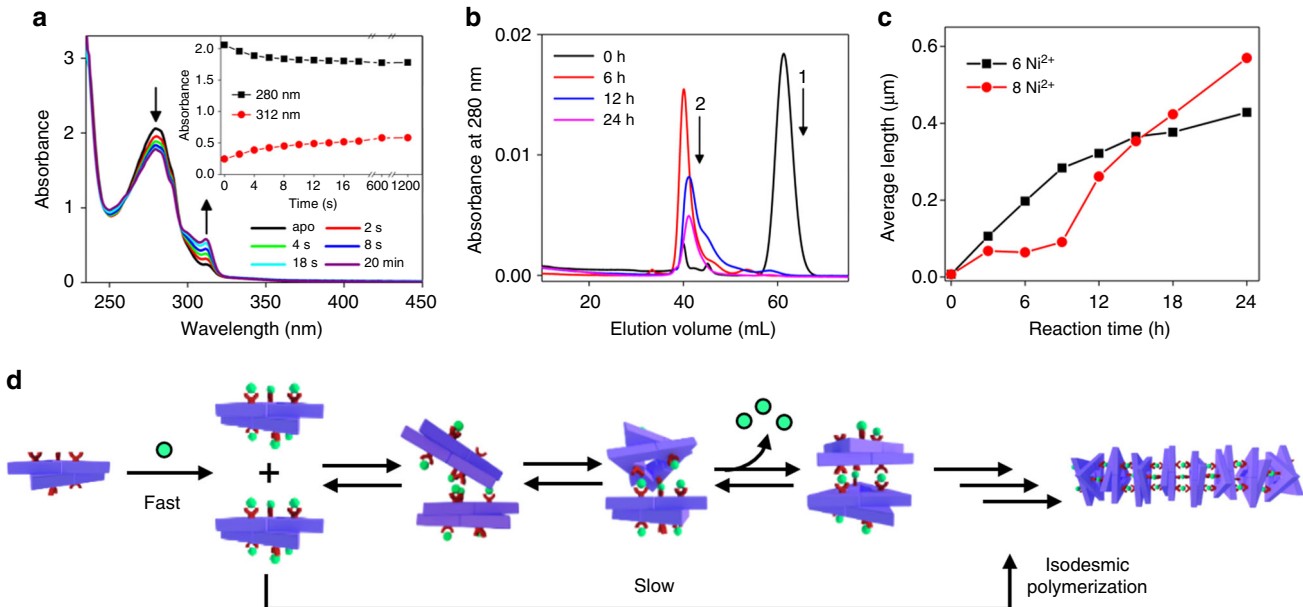

**Fig. 4** Kinetics of 1D-rod formation. **a** Optical spectral changes upon the addition of $Ni^{2+}$ to K44Z at 22 °C. Inset: Time-dependent absorbance at 280 and 312 nm. **b** Elution trace of K44Z incubated with $Ni^{2+}$ (3 equiv.) at 22 °C for 0–24 h from S200 column. The sequential order of the reactions was labeled. **c** Time-dependent rod-growth with K44Z (5 μM) at 37 °C monitored by TEM. **d** A proposed scheme for 1D assembly. The raw data in Fig. 4a, c are provided as a Source Data file.

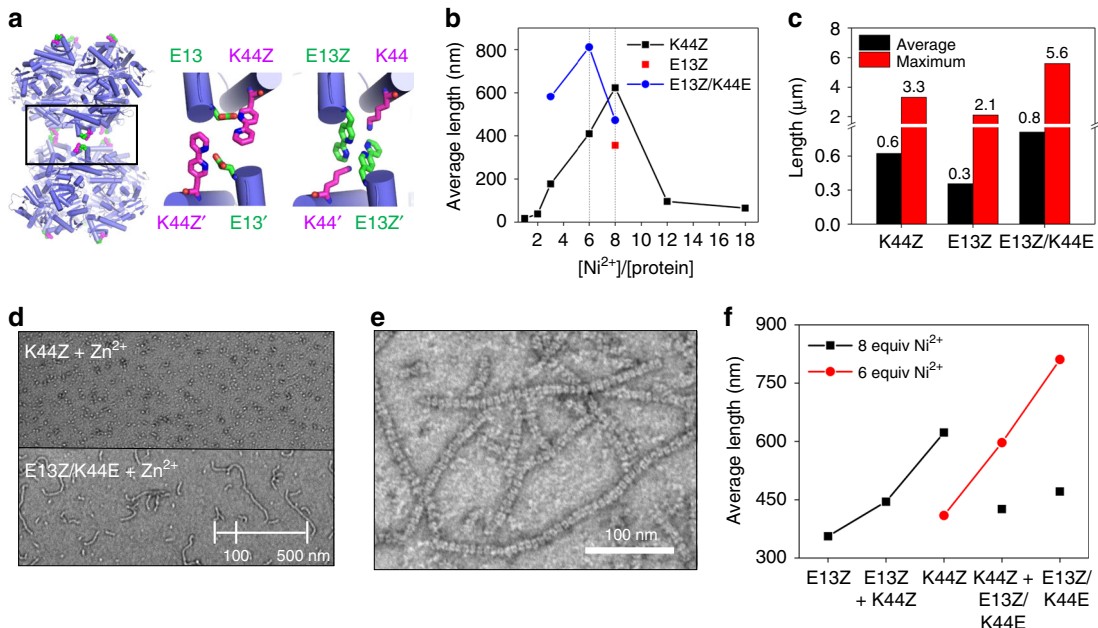

**Fig. 5** Formation of rods with E13Z and E13Z/K44E. **a** Structural models of K44Z and E13Z. **b** Optimization of $Ni^{2+}$ ratios to the protein. **c** The average and maximum lengths from each variant when the ratio of $Ni^{2+}$ to protein is optimized in (**b**). **d** Altered reactivities of E13Z/K44E versus K44Z with 3 equiv. of $Zn^{2+}$. **e** TEM images of pre-mixed K44Z and E13Z/K44E (5 μM each) with 6 equiv. of $Ni^{2+}$ at 37 °C. **f** The average length of rods formed with various variants when 6 (red spheres) or 8 equiv. (black squares) of $Ni^{2+}$ were applied. The raw data in Fig. 5b, c, f are provided as a Source Data file.

mutant yielded linearly assembled products upon the addition of $Ni^{2+}$, and they were even longer than the K44Z-derived ones when the metal to protein ratio was optimized (Fig. 5b, c and Supplementary Figs. 33a, b). Interestingly, the E13Z/K44E mutant generated linearly assembled products with other divalent metal ions, although $Ni^{2+}$ was still the most optimal. The promiscuous inorganic reactivity of the E13Z/K44E variant contrasted with that of K44Z, which formed no linear species with $Zn^{2+}$ (Fig. 5d

and Supplementary Fig. 33c), implying that K44Z and E13Z/K44E mutants exhibit distinct coordination environments for the [M(bpy)₂] species. These data indicate that linear polymerization is driven by metal coordination and is tunable with additional mutations and/or metal substitution.

All three variants, K44Z, E13Z, and E13Z/K44E, provided the end-to-end linkage with $Ni^{2+}$, and 1D self-assembled structures were formed even when the mutants coexisted in solution. When

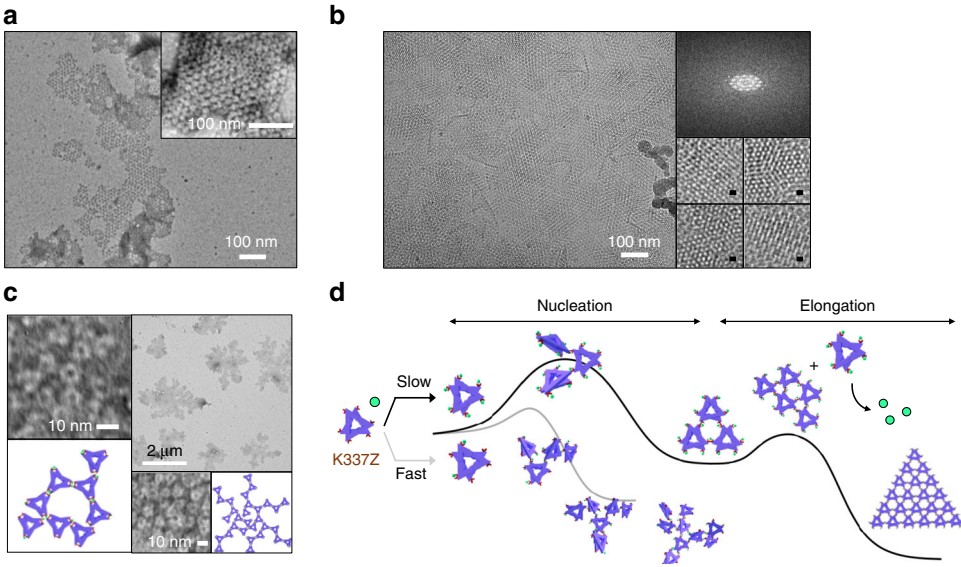

**Fig. 6** Formation of 2D planes. **a** TEM and **b** cryo-TEM images of K337Z (5–10 μM) incubated with 6 equiv. $Ni^{2+}$ at 37 °C for 24 h. **c** TEM images of W298Z (5–10 μM) with 10 equiv. $Ni^{2+}$ at 37 °C (left) or 22 °C (right) for 24 h. **d** A proposed scheme for two-directional protein assembly.

two of the three rod-forming variants were premixed at equal ratios followed by the addition of $Ni^{2+}$, protein rods were formed, which were analogous to those with individual proteins (Fig. 5e and Supplementary Fig. 34). Interestingly, the optimal ratio of $Ni^{2+}$ to the protein varied depending on the mutant (Fig. 5b), and the average and maximum length of the rods varied according to the given metal to protein ratios (Fig. 5f), implying that thermodynamic and kinetic controls are operative even in the mixtures of two discrete variants. These results also indicate that the self-assembly of two discrete proteins is possible with an identical molecular connector module of $[Ni(bpy)_2]$.

**Synthesis of 2D assemblies.** When the bipyridyl groups were anchored to lateral positions such as K337 and W298, the addition of $Ni^{2+}$ formed opaque colloidal and aggregated solutions, respectively. Negative staining TEM and cryo-TEM demonstrate that K337Z forms micron-sized, crystalline, 2D plane-shaped materials in the layer symmetry p312 (Fig. 6a, b and Supplementary Figs. 36 and 37), whereas W298Z yields multibranched fractal-like structures, similar to the kinetically assembled products reported recently (Fig. 6c and Supplementary Figs. 41 and 42)[53]. The preference of K337Z for the layer symmetry p312 rather than the p622 might be associated with their discrete assembly process as follows. Nucleation in p312 required the linkage of two proteins to a single node of $[Ni(bpy)_2]$, and the ensuing protein can bind to the preorganized site cooperatively (Supplementary Fig. 43).

In contrast, the location of bipyridyl groups in W298Z was sterically disabled for p312 assembly, forcing the formation of 2D-assembly in p622. Nucleation in the latter assembly, however, required the linkage of two proteins with two nodes without any structural arrangement for the subsequent protein, resulting in greater complexity and structural defects. As a result, the W298Z protein formed partial p622-like noncrystalline architectures with a substantially greater degrees of irregularities and heterogeneity. Notably, although the driving force for protein self-assembly is identical, two discrete crystalline p312 versus noncrystalline with partial p622 materials were generated depending on the position of the bpy ligand, demonstrating that diverse architectures can be elicited by a single mutation that shifts the position of the bpy ligand.

We further screened the reaction conditions to optimize the formation of the K337Z-derived 2D crystalline products by altering the protein concentration, the metal to protein ratio, and the reaction temperature (Supplementary Fig. 39). Although the stacking interaction between the planes did not allow us to measure the accurate size of individual planes, we identified that desirable assembled structures were formed only under the following conditions; incubation of 5–10 μM K337Z and 6 equiv. of $Ni^{2+}$ at 37 °C for 24 h. Notably, 2D materials were formed only with in this narrow window of conditions, which sharply contrasted to those of the 1D assembly, implying that the kinetic and thermodynamic parameters associated with 2D self-assembly might be counteractive. Kinetic control might be operative given that higher than stoichiometric amounts of $Ni^{2+}$ and less than 10 μM protein are required for the 2D assembly, providing an adequately slow reaction rate for the desirable nucleation step. Concomitantly, sufficiently high temperatures (37 °C) and not less than 5 μM protein may thermodynamically drive self-assembly via bimolecular reactions between $[Ni(bpy)]$ species and $Ni^{2+}$ release.

To estimate the kinetics of the K337Z-derived 2D crystalline material formation, we collected TEM images of the K337Z protein (5 μM) after 18 h of incubation with $Ni^{2+}$ instead of 24 h. Considerable amounts of protein remained unreacted, indicating that the 2D assembly might be accompanied by a lag phase of slow nucleation followed by a propagation step (Fig. 6d and Supplementary Fig. 40). The kinetics of the 2D assembly contrasts with those of the 1D formation. Despite the discrete kinetic process, thermodynamic control associated with NaCl or LiCl was also operative (Supplementary Fig. 38), indicating that the 2D assembly is also thermodynamically driven by the selective formation of $[Ni(bpy)_2]$ nodes.

**Synthesis of combinatory and hierarchical structures.** The $[Ni(bpy)_2]$ formation solely enables protein-assembly and dictates the direction of polymerization. Therefore, we surmised that more complex hybrid structures can be created by mixing 1D and 2D growing variants, where each of the two bpy ligands in $[Ni(bpy)_2]$ is derived from K44Z and K337Z, respectively. K337Z (2D variant) and $Ni^{2+}$ were added to the preformed, short rods derived from K44Z (1D variant), resulting in triangular p312

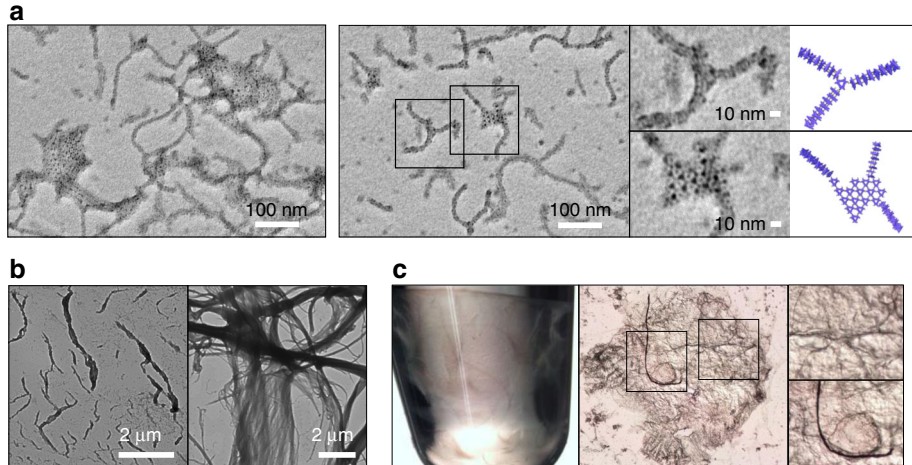

**Fig. 7** Various structures derived from 1D rods and/or 2D planes. **a** Formation of hybrid structures upon the addition of K337Z (2 μM) and 3 equiv. of $Ni^{2+}$ to the pre-formed K44Z-derived rods (10 μM) and 3 equiv. $Ni^{2+}$ at 22 °C for 4 h. TEM images were obtained after 20 h incubation at 22 °C. **b** Addition of uranyl acetate (0.2 wt%) to the K44Z-derived rods. **c** Addition of polar organic solvents such as methanol and acetone to the optimized E13Z/K44E-derived rods (fourfold v/v).

planes cross-linked with the protein rods or multiple rods stretched over a triangular plane (Fig. 7a and Supplementary Fig. 44). The data indicate that their protein–protein interactions are effectively operative even when two variants with different directionalities of polymerization coexist, and the driving force originating from the [Ni(bpy)₂] formation is versatile enough to create diverse structures with only single-site mutations.

In addition, hierarchical structures were synthesized from the 1D assembled linear products (Fig. 7 and Supplementary Fig. 45). The addition of excess uranyl acetate induced electrostatic interactions between the protein rods, further converting them into braided fibrils (Fig. 7b). Analogous transformations were also induced by the addition of polar organic solvents such as methanol and acetone (Fig. 7b, c, Supplementary Fig. 46), possibly increasing the effective interactions between the NaCl salt and protein. The formation of hierarchical architectures was dependent on the applied solvents and the bpy-variants. In particular, the E13Z/K44E rods, but not K44Z rods, formed fibrils upon the addition of the polar organic solvents, indicating that their higher stability, as estimated from their average length, might be related to the concurrent structural changes under the harsh conditions. E13Z/K44E rods form fibril-shaped materials rather than aggregates, indicating that the metal-mediated protein–protein interactions are sufficiently resilient and operative even after the introduction of external chemical agents.

**Chemical properties of the self-assembled structures**. Assembled protein structures may exhibit higher thermal stability, resulting in an effective strategy to transform highly efficient enzymes into more versatile and durable biocatalysts[54]. The optimal temperature for both the 1D- and 2D-assemblies was 37 °C, partially due to the thermal instability of the protein, as described above. Once the assemblies were constructed, however, the 1D and 2D-materials stabilized the protein building blocks even after heating for 24 h at 55 °C by retaining the assembled structures (Fig. 8a and Supplementary Fig. 47). E13Z/K44E rods exhibited no considerable alterations in length before and after heating. In contrast, E13Z- and K44Z-derived rods aggregated or became truncated upon heating, respectively, indicating that the thermal stability correlated with the relative native length of the 1D-rods measured at 37 °C. Notably, after heating, the E13Z/K44E rods were further extended in the presence of greater than

optimal amounts of $Ni^{2+}$ (8 equiv). It is possible that the 1D assembled structures are sufficiently stable to reach the thermodynamic equilibrium at 55 °C when sufficient amounts of $Ni^{2+}$ are present. In addition, no time-dependent change in the shape and length was observed (Supplementary Fig. 48), indicating that the shorter K44Z rods are thermodynamically determined products at 55 °C rather than being gradually degraded during heating.

With the gained thermal stability, we measured the native enzymatic activities of the assembled protein structures (Fig. 8b and Supplementary Fig. 49). The protein selected for assembly in our study is natively an acetyltransferase that catalyzes the transfer of an acetyl group to an amino group of various aminoglycosides such as kanamycin[43]. First, we measured the steady-state specific activities of the bpy-incorporated variants at room temperature in the absence of $Ni^{2+}$. The activities of the K44Z and K337Z variants were comparable to those of the wild-type hexameric protein, whereas the E13Z and E13Z/K44E mutants were substantially inactivated, indicating that the E13Z mutation, but not the K44Z and K337Z mutations might alter the critical steps in catalysis.

Then, we measured the catalytic activities of the K44Z-derived 1D-rods and K337Z-derived 2D-planes at room temperature. Both assembled materials were catalytically active, indicating that the protein-protein interactions created by [Ni(bpy)₂] exerted no considerable perturbation in enzyme catalysis. Subsequently, we measured the catalytic activities of the 1D- or 2D-materials at room temperature after heating at 55 °C. Whereas the wild-type, K44Z, E13Z, E13Z/K44E, and K337Z proteins exhibited no catalytic reactivity after heating at 55 °C, the K44Z-derived 1D and K337Z-derived 2D materials are kinetically competent, yielding specific activities comparable to those of the unheated wild-type protein. Therefore, these data demonstrate that the thermally stabilized protein-assembled materials preserve their native catalytic functions, implying that natural enzymes can be transformed into more versatile and stable biocatalysts.

## Discussion

Genetic incorporation of noncanonical chelating amino acids for site-selective metal coordination endowed dynamic and controllable protein self-assembly. Notably, the sequence of the selected protein was not optimized for self-assembly, and the

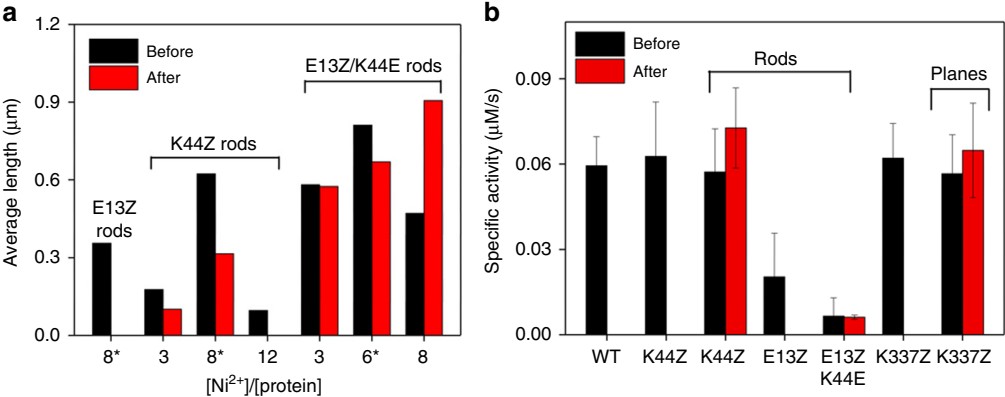

**Fig. 8** Thermal stability and enzymatic activities of protein rods and planes. **a** Changes in the length of rods formed at 37 °C before and after heating at 55 °C. The optimal $Ni^{2+}$ ratios for each variant are shown with asterisks. **b** The native enzyme activities as acetyltransferases in the reactions with acetyl coenzyme A and kanamycin before and after heating at 55 °C. The error bars for the unheated samples (black columns) in (**b**) indicate the standard deviations of the 11, 23, 12, 5, 4, 11, and 13 runs of kinetic measurements with the wild-type, K44Z, K44Z-rods, E13Z-rods, E13Z/K44E-rods, K337Z, and K337Z-planes, respectively. The error bars for the heated samples (red columns) indicate the standard deviations of the 10, 3, and 9 runs of the experiments with K44Z-rods, E13Z/K44E-rods, and K337Z-planes, respectively. The raw data are provided as a Source Data file.

protein is composed of 2328 amino acids, including 90 aspartate, 252 glutamate, and 66 histidine residues, which can interact with metal ions and trigger the formation of undesirable products. Instead, diverse structures ranging from 1D, 2D, hybrid, and fibril structures were synthesized by mixing a single protein, ligand, and metal ion. Thermodynamic and kinetic controls of the reversible inorganic reactivity lead to the formation of diverse architectures with tunability, in terms of the shape, length, and stability while retaining native catalytic activity. Therefore, metal-dependent polymerization may provide effective routes to tailor structurally and functionally versatile protein-based catalysts and materials. Our work demonstrated that 1D assembly is more feasible than 2D assembly, and presumably 3D protein assembly to the formation of well-ordered microcrystals might be even more challenging. Because the formation of $[Ni(bpy)_2]$ species is orthogonal to the introduction of other covalent or noncovalent interactions to elicit protein assemblies, we speculate that the scope of target proteins and their concurrent assembled structures can be significantly expanded. Multidimensional or multi-component protein assemblies can be created, resulting in high levels of structural and functional complexity such as artificial cells or protein compartments for cascade reactions or dynamic motions upon altered chemical environments.

## Methods

**General procedure**. Chemicals or metal salts were purchased from Sigma Aldrich, Acros organics, Alfa Aeser or LPS solution and used without further purification unless described. 1H-NMR spectra were collected with Varian 500 MHz NMR using $CDCl_3$ or $CD_3OD$ solvents. All metal salts were prepared by dissolving metal chloride ($MnCl_2$, $CoCl_2 \cdot 6H_2O$, $NiCl_2 \cdot 6H_2O$, $CuCl_2 \cdot 2H_2O$, $ZnCl_2$, NaCl, LiCl) in $ddH_2O$. All aqueous solutions were prepared with $ddH_2O$ and filtered with 0.22 μm syringe-filters before usage.

**Synthesis of Bpy-Ala amino acids**. The chelating unnatural amino acids were prepared as described previously[55–57], and was briefly summarized below and in Supplementary Fig. 1.

**Synthesis of 2-trimethyltin pyridine (1)**. To a suspension of 2 mm metallic sodium (3.03 g) in dimethoxyethane (28 mL), trimethyltin chloride (8.72 g) dissolved in dimethoxyethane (7 mL) was added dropwise with stirring over 15 min at 0 °C under argon stream. After stirring for 2 h at 0 °C, unreacted sodium was removed from green suspending solution. 2-chloropyridine (3.3 mL) dissolved in dimethoxyethane (21 mL) was added dropwise to green suspending solution at 0 °C under argon stream. The reaction continued for 3 h at 0 °C under argon stream. Dimethoxyethane was removed in vacuum, and the product (**1**) was obtained via

extraction using diethyl ether. The solution (**1**) was used in next step without further purification.

**Synthesis of methyl 2,2′-bipyridine-5-carboxylate (2)**. For coupling reaction, *m*-xylene was purified with molecular sieve prior to distillation. A stirring solution of (**1**), *m*-xylene (100 mL), $Pd(PPh_3)_2Cl_2$ (1.05 g), and 6-chloromethylnicotinate (4.98 g) equipped with reflux condenser was heated to 160 °C. The reaction mixture changed to black color. After 7 h, the reaction mixture was cooled to room temperature and was stirred for 15 min with cellite. After filtration and evaporation, the reaction crude mixture was applied to silica gel flush column chromatography (eluent hexane/ethyl acetate = 3:1 with 5% of triethylamine) to yield the desired product (**2**) (5.2 g, 68% via two steps). 1H-NMR ($CDCl_3$-$d_1$, 500 MHz) of (**2**) (Supplementary Fig. 2): δ 3.99 (s, 3H), 7.37 (s, 1H), 7.86 (t, J = 7.3 Hz, 1H), 8.42 (d, J = 7.9 Hz, 1H), 8.56–8.46 (m, 2H), 8.72 (s, 1H) and 9.28 (s, 1H).

**Synthesis of (2,2′-bipyridin-5-yl) methanol (3)**. Anhydrous tetrahydrofuran was prepared by distillation with sodium-benzophenone after dehydration with molecular sieve. LiAlH_4 (0.76 g) in THF suspension was added dropwise at −78 °C to a stirring solution of (**2**) (4.0 g) in 55 mL of anhydrous THF under argon stream. The resulting solution was slowly warmed to −20 °C, and stirred for 1 h. Then, 10% aqueous THF solution (50 mL) was added very slowly at −78 °C, and warmed to room temperature. Resulting orange solution was treated with celite, stirred for 15 min, and filtered. Red colored oil (**3**) (3.34 g) was extracted with DCM and was used without further purification.

**Synthesis of 5-(bromomethyl) 2,2′-bipyridine (4)**. To a stirring solution of (**3**) in DCM (75 mL), N-bromosuccinimide (3.66 g) and triphenylphosphine (5.18 g) were added at 0 °C, and resulting solution was stirred for 1 h at the same temperature. After concentrating the solution to ~7 mL, silica column chromatography was applied (eluent hexane: ether = 2:1 to 1:1) to yield white solid (**4**) (2.67 g, 60% in two steps). 1H-NMR ($CDCl_3$-$d_1$, 500 MHz) of (**4**) (Supplementary Fig. 3): δ 4.55 (s, 2H), 7.36–7.30 (m, 1H), 7.85 (m, 2H), 8.40 (s, 1H), 8.42 (s, 1H) and 8.69 (s, 2H).

**Synthesis of (2,2′-bipyridin-5-yl)diethyl acetacetomalonate (5)**. To a stirring solution of acetacetomalonate (3.3 g) and sodium ethanolate (1.03 g) in anhydrous EtOH (90 mL), (**4**) (2.5 g) was added and refluxed overnight. After evaporation, the crude reaction product was purified by silica gel flush column chromatography (eluent: hexane: EA = 2:1 to 1:1; then, DCM: MeOH = 15:1 to 10:1) to yield (**5**) (2.7 g, 70%). 1H-NMR ($CDCl_3$-$d_1$, 500 MHz) of (**5**) (Supplementary Fig. 4): δ 1.32 (dt, J = 7.1, 4.4 Hz, 6H), 2.09 (dd, J = 4.6, 2.0 Hz, 3H), 3.75 (s, 2H), 4.22–4.34 (m, 4H), 6.61 (s, 1H), 7.32 (t, 1H), 7.49 (d, J = 7.9 Hz, 1H), 7.82 (t, J = 7.6 Hz, 1H), 8.33 (dd, J = 20.6, 7.5 Hz, 3H) and 8.68 (s, 1H).

**Synthesis of (2,2′-bipyridin-5-yl)alanine or Bpy-Ala (6)**. (2,2′-bipyridin-5-yl) diethyl acetacetomalonate (**5**) (2.7 g) in 37% HCl (75 mL) was heated to reflux overnight. Removing of HCl lead to formation of final product, Bpy-Ala. It was used for incorporation of unnatural amino acid to the selected proteins without further purification. 1H-NMR (MeOD-$d_1$, 500 MHz) (Supplementary Fig. 5): δ 3.49 (ddd, J = 41.0, 14.6, 6.7 Hz, 2H), 3.78 (s, 2H), 4.49 (t, J = 6.7 Hz, 1H),

8.08–8.02 (m, 1H), 8.29 (d, $J = 8.2$ Hz, 1H), 8.56 (d, $J = 8.3$ Hz, 1H), 8.64 (t, $J = 7.9$ Hz, 1H), 8.76 (d, $J = 8.2$ Hz, 1H) and 8.94–8.87 (m, 2H).

**The selection of divalent metal ions for the current work**. The related thermodynamic parameters of the first-row divalent transition metal ions with bipyridine ligands were included in Supplementary Table 1. We considered following metal binding equilibrium for selection of metal ions. Equilibrium constants of the Supplementary Eqs. (1)–(3) in Supplementary Note 1 were listed in Supplementary Table 1.

**Incorporation of bpy-Ala to the selected proteins**. The sequence of the selected protein (PDB code 3N7Z) was obtained from RCSB website and included in Supplementary Table 2. The gene fragment was synthesized after codon optimization for *Escherichia coli* heterologous expression (Gene Universal Inc.) and incorporated into pET-28(a) vector using Nde1 and Xho1 restriction cut-sites. All PCRs were conducted using KOD plus neo polymerase kit purchased from Toyobo. Amber codon was used for the incorporation of Bpy-Ala to the selected residue using the custom-designed primers (Supplementary Table 3). The plasmid for Bpy-Ala incorporation, pEVOL, was a kind gift from prof. Hak Joong Kim at Korea University. Plasmids were extracted by using DNA purification kit (Labopass). Custom-designed primer synthesis and plasmid sequencing were carried out in Macrogen. For gene cloning and heterologous protein expression, DH5α and BL21 (DE3) *E. coli* strains (NEB) were used, respectively.

For protein expression, plasmids were transformed to BL21(DE3) *E. coli* competent cells (NEB), containing pEVOL plasmid for the incorporation of Bpy-Ala[47]. A single colony grown in the LB/agar plate containing kanamycin/chloramphenicol were inoculated in TB culture (20 mL) containing kanamycin and chloramphenicol at 37 °C. After 9 h, the cell cultures (7.5 mL) were added to 0.75 L TB media containing kanamycin (50 mg/mL) and chloramphenicol (35 mg/L) at 37 °C and were grown in orbital shaker, 170 rpm at 37 °C for 4 h. When OD$_{600}$ reached ~1.0–1.2, 1 mM Isopropyl β-D-1-thiogalactopyranoside (IPTG), 1 mM Bpy-Ala, and 0.2% arabinose were added at final concentrations at 30 °C. After 12 h, cell pellets were harvested by centrifugation at 4715×g, 4 °C and kept at −80 °C. Incorporation of the chelating amino acid and expression of full-length proteins was confirmed by sodium dodecyl sulfate-poly acrylamide gel electrophoresis (SDS-PAGE) (Supplementary Fig. 6a).

**Purification of the selected proteins**. The harvested cells from 1.5 L of TB media were resuspended in lysis buffer (pH 8.0, [HEPES] = 50 mM, [NaCl] = 500 mM, [2-mercaptoethanol] = 10 mM, [Imidazole] = 5 mM, 10% w/v glycerol) at 4 °C, sonicated in iced bath (on/off = 1.0 s/1.5 s for 30 min). After centrifugation at 18,800×g at 4 °C for 35 min and syringe filtration, the soluble fraction was directly applied to 5 mL His Trap-FF (GE Healthcare Life Sciences) with ÄKTA pure protein purification system at 4 °C (Supplementary Fig. 6b). Eluted samples were treated by thrombin with 20 mM EDTA at 22 °C overnight to delete the His-6 tag prior to the N-terminus. The resulting samples were purified by His Trap-FF and Superdex 200 gel filtration columns (GE Healthcare Life Sciences) with the following buffer, [Tris-HCl] = 50 mM, [NaCl] = 150 mM, pH 8.0, at 4 °C (Supplementary Fig. 6c). Then, the samples were used for the reactions with metal ions after buffer exchange with salt-free buffer. All purified proteins were characterized by SDS-PAGE (Supplementary Fig. 6d). Protein concentration was determined by measuring the absorbance at 280 nm (UV–vis spectrophotometer, Agilent Technologies Cary 8454 UV–vis) by using the epsilon values of protein and bipyridine at 280 nm[58].

**Sample preparations**. Reactions of the bpy-variants with Ni$^{2+}$ were carried out with 50 mM Tris, 150 mM NaCl, pH 7.1 buffer at 37 °C for 24 h. It is noteworthy that depending on the temperature, the pH value of the Tris buffer changes as 8.0, 7.5, 7.1 at 4, 22, 37 °C, respectively[48]. To monitor pH-dependent formation of protein-assembly, 50 mM MES pH 6, 50 mM Tris at pH 7–8, 50 mM CHES pH 9 buffers were prepared and applied at 37 °C.

**Transmission electron microscopy**. TEM grid and uranyl acetate were purchased from Electron Microscopy Sciences. Carbon-coated 200-mesh copper grids (CF200-Cu) were applied to glow discharge using PELCO easy glow (Ted Pella Inc.). For nanorods, each sample was diluted with the buffer used for sample preparation to 0.05 μM at final concentration just before loaded to TEM grid. For 2D-planes, the sample was loaded without dilution. After incubation for 1 min, any remaining liquid was removed by filter paper, washed with ddH$_2$O, and stained with 1 wt% uranyl acetate solution. After drying, TEM was operated at HITACHI H–7600 (HITACHI–Science & technology, 120 kV), using a software AmtV542. The focal point was set in a manner that the rods were seemingly a hollow tube-like material because it enables us to distinguish singly aligned tubes from interweaved multi-stranded fibrils (Supplementary Fig. 7).

TEM images were analyzed using an image analysis software, Digimizer. The lengths of at least 100 protein-rods were measured to estimate the number average

length ($L_n$), weight average length ($L_w$), and PDI for each condition as

$$L_n = \frac{\sum_{i=1}^{N} N_i L_i}{\sum_{i=1}^{N} N_i}, \tag{1}$$

$$L_w = \frac{\sum_{i=1}^{N} N_i L_i^2}{\sum_{i=1}^{N} N_i L_i}, \tag{2}$$

$$\mathrm{PDI} = L_w / L_n, \tag{3}$$

where $N$ is the total number of rod-species examined from TEM images, $L_i$ is the length of each rod, and $N_i$ is the number of species of which length is $L_i$. To examine the effects of reactions conditions on the 1D-rod growths, we plotted the number average length ($L_n$) values as a function of metal to protein ratios, temperatures, and protein concentrations. To depict the length distributions of the 1D-rods, the measured length values were binned into 50 nm ranges and the frequency of each range was plotted into the bar charts.

**UV–vis spectroscopy**. The Ni$^{2+}$ binding to bpy-anchored proteins were monitored by UV–vis spectrophotometer (Agilent Technologies Cary 8454 UV–VIS). To detect the time-dependent spectral changes, 6 equiv of Ni$^{2+}$ were add to protein (5 μM) in 50 mM Tris-HCl, 150 mM NaCl, 400 μL at final volume at 25 °C.

**Size exclusion chromatography**. The oligomeric states of the assembled proteins were determined by size exclusion chromatography using Superdex 200 gel filtration columns (GE Life Sciences) ([Tris-HCl] = 50 mM, [NaCl] = 150 mM, pH 8.0 buffer at 4 °C) with ÄKTA pure protein purification system at 4 °C.

**Cryo-TEM**. Protein sample (3 μL, 5–10 μM K337Z) reacted with metal ions were loaded on glow-discharged quantifoil (R 2/2, 400 mesh holey-carbon grid, Quantifoil Micro Tools), blotted for 2 s at 4 °C and frozen using Vitrobot (Thermo Fisher Scientific, USA) in liquid ethane. The samples were analyzed using Talos L120C TEM (Thermo Fisher Scientific, gun type: Lab6, 120 kV) at National Instrumentation Center for Environmental Management (NICEM) at Seoul National University.

**Atomic force microscope (AFM)**. For acquiring AFM image, total 50 μL of pre-made rods (10 μM) were prepared and buffer was exchanged to ddH$_2$O to eliminate remaining salts using centrifugal minispin-column (Merck Millipore). Diluted samples were loaded to piranha solution treated silicon wafer with SiO$_2$ thin layer and dried in the air. AFM images were acquired with park systems XE-70 AFM using an AFM tip non-contact cantilever (PPP-NCHR 10M, Park SYSTEMS) with tapping mode.

**Catalytic activity assays**. Natively, the selected protein is an aminoglycoside acetyltransferase, which acetylate amines of aminoglycosides, such as kanamycin upon the reaction with acetyl coenzyme A, acetyl-S-CoA. As a result, CoA-SH is generated as a product, of which concentrations can be quantitatively determined by the addition of Ellman's reagent with the absorption changes at 412 nm (Supplementary Fig. 49)[43]. We used 96-well plate reader (SYNERGY H1 microplate reader, BioTek) to monitor the absorption changes at 412 nm at the intervals of 15 s for 10–15 min upon the addition of acetyl CoA (0.5 mM at final concentration) to the solution ([Tris-HCl] = 50 mM, pH 8.0 buffer at 4 °C) containing proteins or protein-assembled structures (0.5 μM hexamer at final concentration), kanamycin (KAN) (0.5 mM), and Ellman's reagents (2 mM) at 25 °C.

**Reporting summary**. Further information on research design is available in the Nature Research Reporting Summary linked to this article.

## Data availability
Data are available from the corresponding authors upon reasonable request. Representative data for this article is available as a Supplementary Information file. The source data underlying Figs. 2c, 3, 4a, c, 5b, c, f, 8 and Supplementary Figs. 6a, d, 8, 9b, 12b, 14, 16, 17, 20, 22, 24, 25, 29, 31, 32b, 33b, 34b, and 48 are available in a Source Data file.

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

## Acknowledgements

This work was supported by the National Research Foundation (NRF) grant (NRF-2019R1C1C1003863). The authors are also grateful to Prof. Byeong-Hyeok Sohn for allowing us to use a TEM instrument, Prof. Jungwon Park and Dr. Min-ho Kang for the helpful assistance in glow discharge experiments, and Prof. Zee Hwan Kim and Mr. Gyou Il Jung for the guidance in AFM experiments.

## Author contributions

M.Y. and W.J.S. designed the project, M.Y. performed the experiments, and M.Y. and W.J.S. wrote the paper.

## Competing interests

The authors declare no competing interests.
