## [Peer Review File · Nature Communications]

REVIEWERS' COMMENTS:

Reviewer #1 (Remarks to the Author):

Metal-directed self assembly of peptides and proteins is a wide-ranging field, and many successful examples have been reported of complex materials that take advantage of some combination of biomolecular folding and metal coordination. These precedents can be divided into two broad categories: (1) systems that make use of canonical amino acid side chains exclusively as the metal-binding groups and (2) systems that use ligand moieties not found in natural proteins to this end. The use of non-biological ligands expands the scope of assemblies possible but also raises the challenge of how to incorporate such groups in natural host sequences. The solution to this challenge in prior work has been to either prepare the desired construct by total chemical synthesis (limiting the size of the chain accessible) or carry out chemical modification of a natural side chain in biologically produced material (limiting the number and types of artificial groups that can be introduced). The central innovation in the manuscript under consideration is to demonstrate unnatural amino acid mutagenesis as a useful third approach to produce complex materials from expressed proteins bearing non-biological metal coordinating side chains. Yang and Song base their design on an acetyltransferase from *B. anthracis* as a building block for the materials. Of note is its complex quaternary structure, consisting of a hexameric assembly with D₃ symmetry. They introduce non-natural metal-binding moieties at defined sites in the hexamer through amber suppression methods using a known expression system for introduction of a bipyridine containing analog of phenylalanine. Based on the position of the introduced metal-binding mutation and control of assembly conditions, they are able to produce 1-dimensional fibers and 2-dimensional lattices in the presence of Ni²⁺. The 2-dimensional lattices and associated characterization data, in particular, are quite striking and provide compelling evidence for the rational control over material morphology through fine control over metal binding residue placement. Importantly, the authors show the enzymatic activity of the protein building block is retained in the metalloprotein assembly context, and the 2-dimensional lattice imparts stability to inactivation by high temperature. Overall, this is a strong manuscript. The writing is clear for the most part and the scientific findings novel and significant. Publication in *Nature Communications* is recommended, pending revision.

- The authors should add a reference on the report of the use of the related bidentate aromatic nitrogen ligand phenanthroline to direct protein assembly [DOI: 10.1021/ja9000695].

-Figure 2b needs a scale bar.

Reviewer #2 (Remarks to the Author):

This paper reports on a new approach for driving reversible protein assembly by genetically installing a bipyridine residue into an oligomeric (D₃) protein. Experiments show that Ni ions selectively drive coordination of two bipyridine groups. Inserting the bipyridine at the top/bottom surface of the hexameric protein gave rod-shaped filaments. Inserting the bipyridine around the perimeter gave rise to 2D assemblies. Layer symmetries p312 and p622 were observed for different positions of the bipyridine. The design idea is new and the demonstration of utility – i.e. successful formation of extended assemblies – is compelling. This should make an important addition to the literature in the area of designing protein assemblies. [As an aside I note that the paper shows impressive breadth and expertise for a single-junior-author effort.] I have several comments and concerns that I think could be address relatively easily prior to publication.

1) The paper would be improved by abbreviating the long sections on conditions for optimizing (1D) filaments and their kinetics. It is admittedly important that the assembly outcomes turn out to depend on mutation position, solution conditions, concentrations, times, etc. But the essential ideas -- that the outcomes vary and the conditions are likely to require optimization -- could be stated much more succinctly. The highly specific details observed for this enzyme are not likely to translate directly to other proteins, so that information could be relegated to the supplement or otherwise condensed.

2) Throughout the paper the authors describe their 2D assemblies in terms of "space groups" and they give space group designations (P312 and P622). This is incorrect. 2D materials need to be described as conforming to "layer groups" (of which there happen to be 17 possibilities). The layer groups are designated by lower case leading letters for the lattice type: the authors mean p312 and p622. The authors need to make the appropriate changes throughout: changing space group to layer group, and using the correct lower case designations.

3) The authors find that forming effectively 1D rods is reasonably easy and robust, whereas forming well-ordered 2D arrays seems to be somewhat harder, i.e. having a smaller window of success (one of the layer symmetries seems to be easier to make than the other possibility). The authors do not mention that (according to Yeates, 2016. Curr Opin) there are six different 3D crystalline materials possible by combining a D3 hexamer with a 2-fold symmetry (form the bidentate metal+ligand). Some of these would presumably be geometrically permissible for the protein being investigated, but it seems that well-ordered 3D (micro) crystals are not obtained in any of the experiments. It would add a further interesting point if the authors were to note this trend if facility: 1D > 2D > 3D. Indeed, reliably designing extended/crystalline 3-D protein materials in the same way that cages (and some layers) have been designed has remained mostly elusive so far.

4) In the legend to Figure 6, the attempt to articulate a kinetic ordering (1-4) in part d isn't very clear and doesn't add much of substance that isn't either self-evident or speculative. It would be better to remove this.

5) In one place (line 287) the authors describe their rods/filaments as amyloid-like. This is not the best description. Prevailing definitions of amyloid are quite specific, focusing in particular on the presence of cross-beta form, which is not what is being produced in this study. To avoid confusion, this description should be avoided.

Reviewer #3 (Remarks to the Author):

This is an exciting communication reporting on the use on bpyAla, an unnatural aminoacid, for the metal-mediated assembly of proteins. This field, initiated by Schultz (ref 26) has been thoroughly investigated by Tezcan, albeit with natural Lewis basic aminoacids. Accordingly, the manuscript by Prof. Song is a great addition to these efforts.

The choice of 1)protein scaffold 2) position of the bpyAla and 3) metal is well justified. The non-trivial characterization of the assemblies is of the highest standards.

As taught in basic inorganic chemistry, the chelate effect allows to increase the binding constant, compared to two monodentate donors. The author use this argument to justify the choice. It is unclear to the referee how much the chelate effect is indeed an asset, compared to carefully positioned monodentate ligands provided by natural aminoacids (His, Met, Cys, Glu etc). The referee does not see any advantage over Tezcan's remarkable assemblies.

It is thus suggest that the author spends a few words on the future assemblies that would not be accessible using natural aminoacids. The referee suspects that the introduction of the unnatural aminoacid leads to a dramatic drop in expression level and that the introduction of bpyAla does not lead to protein expression in many cases.

Finally, the referee suggests to include a few references of the group of T. Hayashi on myoglobin assemblies etc.

REVIEWERS' COMMENTS:

Reviewer #1

- The authors should add a reference on the report of the use of the related bidentate aromatic nitrogen ligand phenanthroline to direct protein assembly [DOI: 10.1021/ja9000695].

We appreciate for the suggestion, and we cited the reference 35 in the revised manuscript as an example of using metal coordination for protein self-assembly.

-Figure 2b needs a scale bar.

We added a scale bar in Figure 2b in the revised manuscript.

Reviewer #2

1) The paper would be improved by abbreviating the long sections on conditions for optimizing (1D) filaments and their kinetics. It is admittedly important that the assembly outcomes turn out to depend on mutation position, solution conditions, concentrations, times, etc. But the essential ideas -- that the outcomes vary and the conditions are likely to require optimization -- could be stated much more succinctly. The highly specific details observed for this enzyme are not likely to translate directly to other proteins, so that information could be relegated to the supplement or otherwise condensed.

We appreciate for the reviewer's comment and we agree that the exact condition demonstrated in this work is unlikely to be operative in other proteins. But we described the optimized condition and the concurrent outcomes because the results demonstrate that desirable self-assembled products can be formed as the orchestration of both thermodynamic and kinetic parameters. Therefore, how these chemical factors are related to each other is likely to be still effective for the self-assembly of other proteins.

2) Throughout the paper the authors describe their 2D assemblies in terms of "space groups" and they give space group designations (P312 and P622). This is incorrect. 2D materials need to be described as conforming to "layer groups" (of which there happen to be 17 possibilities). The layer groups are designated by lower case leading letters for the lattice type: the authors mean p312 and p622. The authors need to make the appropriate changes throughout: changing space group to layer group, and using the correct lower case designations.

We appreciate for the reviewer's comment, and we followed the reviewer's suggestion by altering the terms into 2D-layer symmetry or layer groups. The corrections are highlighted in the revised manuscript.

3) The authors find that forming effectively 1D rods is reasonably easy and robust, whereas forming well-ordered 2D arrays seems to be somewhat harder, i.e. having a smaller window of success (one of the layer symmetries seems to be easier to make than the other possibility). The authors do not mention that (according to Yeates, 2016. Curr Opin) there are six different 3D crystalline materials possible by combining a D3 hexamer with a 2-fold symmetry (from the bidentate metal+ligand). Some of these would presumably be geometrically permissible for the protein being investigated, but it

seems that well-ordered 3D (micro) crystals are not obtained in any of the experiments. It would add a further interesting point if the authors were to note this trend if facility: 1D > 2D > 3D. Indeed, reliably designing extended/crystalline 3-D protein materials in the same way that cages (and some layers) have been designed has remained mostly elusive so far.

We agree with the reviewer's comment on the possibilities and challenges in the design of 3-D protein materials. We include the following sentence in the discussion to read, **"Our work demonstrated that 1D-assembly is more feasible than 2D-assembly, and presumably 3D protein-assembly to the formation of well-ordered microcrystals might be even more challenging. Because the formation of [Ni(bpy)₂] species is orthogonal to the introduction of other covalent or non-covalent interactions to elicit protein-assemblies, we speculate that the scope of target proteins and concurrent assembled structures can be significantly expanded. Multi-dimensional or multicomponent protein-assemblies can be created, resulting in the high levels of structural and functional complexity such as artificial cells or protein compartments for cascade reactions or dynamic motions upon altered chemical environments."**

4) In the legend to Figure 6, the attempt to articulate a kinetic ordering (1-4) in part d isn't very clear and doesn't add much of substance that isn't either self-evident or speculative. It would be better to remove this.

As suggested by the reviewer, we moved the figure into Supplementary Figure 43.

5) In one place (line 287) the authors describe their rods/filaments as amyloid-like. This is not the best description. Prevailing definitions of amyloid are quite specific, focusing in particular on the presence of cross-beta form, which is not what is being produced in this study. To avoid confusion, this description should be avoided.

We agree with the reviewer's comment that the term, amyloid, is unnecessarily specific and can be confusing. We deleted the word in the revised manuscript to read, **".... further converting into braided fibrils."**

Reviewer #3

1. As taught in basic inorganic chemistry, the chelate effect allows to increase the binding constant, compared to two monodentate donors. The author use this argument to justify the choice. It is unclear to the referee how much the chelate effect is indeed an asset, compared to carefully positioned monodentate ligands provided by natural amino acids (His, Met, Cys, Glu etc). The referee does not see any advantage over Tezcan's remarkable assemblies. It is thus suggest that the author spends a few words on the future assemblies that would not be accessible using natural amino acids.

As mentioned by the reviewer, Tezcan group have pioneered and demonstrated that metal ions can create protein self-assembly. Building-block proteins used for self-assembly, however, require very well-defined metal-binding residues that are either created by sequence optimization or assisted by pre-existing protein-protein interactions. Otherwise, surface-exposed metal-ligating residues would perturb the desirable reactions, resulting in the formation of non-crystalline protein precipitants. In particular,

the protein selected for our study is composed of 90 aspartate, 252 glutamate, 66 histidine residues and they are potential inhibitor for selective metal-coordination. Therefore, introduction of selective and strong chelating metal-binding ligands are essential to override any potential inhibitory reactions with the natural monodentate ligand. To emphasize the chelation effect in the design of protein self-assembly, we added the following sentences in the discussion section to read,

“Notably, the sequence of the selected protein was not optimized for self-assembly, and the protein is composed of 2328 amino acids, including 90 aspartate, 252 glutamate, and 66 histidine residues, which can interact with metal ions and trigger the formation of undesirable products. Instead, diverse structures ranging from 1D, 2D, hybrid, and fibril structures have been synthesized”

We also speculate that more diverse and complexed structures can be generated by applying the method demonstrated in this work, and we included the following sentences in the Discussion section, to read,

“Multi-dimensional or multicomponent protein-assemblies can be created, resulting in the high levels of structural and functional complexity such as artificial cells or protein compartments for cascade reactions or dynamic motions upon altered chemical environments.”

2. The referee suspects that the introduction of the unnatural amino acid leads to a dramatic drop in expression level and that the introduction of bpy-Ala does not lead to protein expression in many cases.

We agree with the reviewer’s comment on the usage of unnatural amino acids, and we in fact reported in the Supplementary information that we failed to obtain 7 mutants (N299Z, N300Z, V301Z, H326Z, E332Z, E333Z, and D336Z) in large scales.

3. Finally, the referee suggests to include a few references of the group of T. Hayashi on myoglobin assemblies etc.

We have cited a paper from T. Hayashi on myoglobin assemblies in the previous manuscript (reference 25), but we agree that his achievements in this field can be further appreciated. We appreciate for the reviewer’s suggestion, and now we added reference 29 in the revised manuscript.